# Stroke vs. Preeclampsia: Dangerous Liaisons of Hypertension and Pregnancy

**DOI:** 10.3390/medicina59101707

**Published:** 2023-09-24

**Authors:** Milan Lackovic, Dejan Nikolic, Milena Jankovic, Marija Rovcanin, Sladjana Mihajlovic

**Affiliations:** 1University Hospital “Dragisa Misovic”, Heroja Milana Tepica 1, 11000 Belgrade, Serbia; lackovic011@gmail.com (M.L.); mihajlovicobg@gmail.com (S.M.); 2Faculty of Medicine, University of Belgrade, 11000 Belgrade, Serbia; denikol27@gmail.com; 3Department of Physical Medicine and Rehabilitation, University Children’s Hospital, 11000 Belgrade, Serbia; 4Neurology Clinic, University Clinical Center of Serbia, 11000 Belgrade, Serbia; 5Clinic for Gynecology and Obstetrics “Narodni Front”, 11000 Belgrade, Serbia; marija.rovcanin@gakfront.org

**Keywords:** stroke, pregnancy, hypertensive disorder of pregnancy, preeclampsia

## Abstract

Stroke during pregnancy and preeclampsia are two distinct but interrelated medical conditions, sharing a common denominator—blood control failure. Along with cardiovascular diseases, diabetes, dyslipidemia, and hypercoagulability, hypertension is undoubtedly a major risk factor associated with stroke. Even though men have higher age-specific stroke rates, women are facing higher life-long stroke risk, primarily due to longer life expectancy. Sex hormones, especially estrogen and testosterone, seem to play a key link in the chain of blood pressure control differences between the genders. Women affected with stroke are more susceptible to experience some atypical stroke manifestations, which might eventually lead to delayed diagnosis establishment, and result in higher morbidity and mortality rates in the population of women. Preeclampsia is a part of hypertensive disorder of pregnancy spectrum, and it is common knowledge that women with a positive history of preeclampsia are at increased stroke risk during their lifetime. Preeclampsia and stroke display similar pathophysiological patterns, including hypertension, endothelial dysfunction, dyslipidemia, hypercoagulability, and cerebral vasomotor reactivity abnormalities. High-risk pregnancies carrying the burden of hypertensive disorder of pregnancy have up to a six-fold higher chance of suffering from stroke. Resemblance shared between placental and cerebral vascular changes, adaptations, and sophisticated auto-regulatory mechanisms are not merely coincidental, but they reflect distinctive and complex cardiovascular performances occurring in the maternal circulatory system during pregnancy. Placental and cerebral malperfusion appears to be in the midline of both of these conditions; placental malperfusion eventually leads to preeclampsia, and cerebral to stoke. Suboptimal performances of the cardiovascular system are proposed as a primary cause of uteroplacental malperfusion. Placental dysfunction is therefore designated as a secondary condition, initiated by the primary disturbances of the cardiovascular system, rather than an immunological disorder associated with abnormal trophoblast invasion. In most cases, with properly and timely applied measures of prevention, stroke is predictable, and preeclampsia is a controllable condition. Understanding the differences between preeclampsia and stroke in pregnancy is vital for healthcare providers to enhance their clinical decision-making strategies, improve patient care, and promote positive maternal and pregnancy outcomes. Management approaches for preeclampsia and stroke require a multidisciplinary approach involving obstetricians, neurologists, and other healthcare professionals.

## 1. Introduction

Stroke during pregnancy and preeclampsia are two distinct but interrelated medical conditions that pose significant risks to maternal and fetal health. Both conditions can lead to adverse outcomes and have been the focus of extensive research efforts. Stroke in pregnancy, although relatively rare, represents a critical medical emergency with potentially devastating consequences for both the mother and the unborn child. On the other hand, preeclampsia, characterized by hypertension and organ dysfunction, remains one of the leading causes of maternal morbidity and mortality worldwide. While both conditions share some similarities in terms of their impact on maternal well-being, they also exhibit some distinct clinical manifestations [1,2,3].

Understanding the differences in clinical presentation between stroke in pregnancy and preeclampsia is crucial for accurate diagnosis, prompt intervention, and effective management. Such knowledge can help healthcare providers differentiate between these two conditions, leading to improved patient outcomes and tailored treatment strategies. Moreover, unraveling the unique features of each condition may shed light on the underlying pathophysiological mechanisms involved, allowing for targeted interventions and preventive measures.

This comprehensive review aims to provide a synthesis of the existing literature on the clinical manifestation and differences between stroke in pregnancy and preeclampsia. By examining a wide range of studies, we aim to identify key differentiating factors such as the onset, symptomatology, and risk factors associated with these two conditions. Through an in-depth analysis of the available evidence, we will strive to elucidate the distinguishing features of stroke in pregnancy and preeclampsia, facilitating their early recognition and appropriate management.

The review will commence with an overview of the epidemiological aspects of stroke in pregnancy and preeclampsia, highlighting the incidence rates, demographic patterns, and associated risk factors. Subsequently, we will delve into the clinical presentation of each condition, discussing the specific symptoms, neurological deficits, and other relevant diagnostic markers. We will explore the challenges faced in differentiating between stroke in pregnancy and preeclampsia, particularly considering the overlap of certain clinical features.

Overall, this review aims to consolidate current knowledge and provide a critical analysis of the clinical manifestation differences between stroke in pregnancy and preeclampsia. By synthesizing the available evidence, we hope to contribute to a deeper understanding of these complex conditions, ultimately facilitating earlier diagnosis, effective management, and improved outcomes for pregnant women at risk.

## 2. Stroke: Incidence, Risk Factors, Clinical Manifestations, and Gender Balance

Stroke is globally recognized as the second leading cause of death [4], and it is estimated to remain so at least until 2030 [5]. It is associated with impaired vascular perfusion through brain’s blood vessels, and according to pathophysiological mechanisms, there are two main types of stroke—ischemic and hemorrhagic. Hemorrhagic stroke is caused by intracranial bleeding, while deficiency in blood and oxygen supply leads to ischemic stroke. In the general population, ischemic stroke is accountable for approximately 85% of strokes, while 10–15% of cases are associated with hemorrhagic stroke [6].

After prolonged argumentation and advocacy, the newly released International Classification of Disease 11 (ICD-11) has separated stroke from the cardiovascular group of disease, and stroke was rightfully re-categorized as neurological entity [7].

Incidence of stroke is closely related to the prevalence of major, mainly modifiable risk factors, including hypertension, obesity, diabetes mellitus, metabolic disorders, and tobacco consumption [8]. Ageing accounts as the most important predictor of stroke incidence, and along with gender, it belongs to the group of non-modifiable risk factors [9]. Management of modifiable risk factors should therefore be the core tenet of all our effort dedicated to minimization, or at least reduction, of this major global cause of death and disability.

The new millennium has seen a significant decrease in stroke incidence, but unfortunately geographical distribution of this decrease was not universal across the globe. Consequently, we are faced today with significant discrepancy in stroke incidence, dependable on gender, race, ethnicity, and geographical distribution [10,11]. A variation in fatality rates associated with stroke, as well as overall morbidity burden, is notable between high-, middle-, and low-income countries [12], emphasizing the magnitude of the growing inequity in resources between different parts of the world. Between 1990 and 2016, stroke incidence has decreased for 42% in high income countries; unfortunately, during the same time period, this incidence has nearly doubled in middle- and low-income countries [13].

Cognitive and physical impairments following stroke are inevitably affecting patients’ quality of life significantly [14], and with an ageing population, predominately in the high-income countries, sequels following stroke are becoming a rising concern and challenge for healthcare professionals. Projections coming from the European Union estimate a 27% increase among stroke survivors between 2017 and 2047, primarily due to the improved survival rates and ageing population in developed countries [15].

Even though males have higher age-specific stroke rates, females are at higher jeopardy of morbidity and mortality related to stroke, primarily due to longer life expectancy and higher stroke incidence associated with older age [16]. Although stroke incidence is closely related to ageing, females of younger age are affected in higher proportions compared to men [17]. These differences are primarily associated with pregnancy-related complication, use of hormonal therapy and contraception, as well as migraine with aura [17].

Stroke manifestations differ between females and males, and females seem to be more sensitive to stroke. On animal models, Bushnell et al. have demonstrated that two X chromosomes are associated with higher neurological defects, larger infarcts, and immune cells infiltration [18]. On the other hand, hormonal status differences between the genders seem to have a favorable role in outcome prediction. Some of the estrogens protective roles are by now well known, such as blood vessel dilatation, blood flow improvement, and induction of anti-inflammatory factors [19].

Systemic review identified pregnancy-related complications, including hypertensive disorder of pregnancy, stillbirth, and preterm delivery as female-specific risk factors for stroke; the same review identified oophorectomy as risk factor, and controversially the possibly protective role of hysterectomy [20].

There are few significant inconsistencies in clinical presentation and manifestation of acute stroke between the genders, and these differences might pose an important obstacle in timely diagnosis and treatment of acute stroke [21]. Stroke is commonly manifested as limb and facial weakness, paresthesia, dysarthria, and dysphagia, and these symptoms are usually classified as traditional symptoms of disease [22]. Females affected with stroke are more susceptible to experience incontinence, loss of consciousness, swallowing difficulties, and coma [23,24]. This untypical clinical presentation may therefore lead to a delay in making a proper diagnosis and eventually result in significant morbidity and mortality in females.

## 3. Arterial Hypertension and Gender Inequity Distribution

High arterial blood pressure, or arterial hypertension, is the most important contributor of the global burden of disease. Despite being a modifiable risk factor for cardiovascular disease, due to poor disease control, hypertension remains the leading global cause of morbidity and mortality among modern-age men and women [25]. Hypertension is usually closely associated with older age, but unfortunately it affects all age groups, and it is major contributor of disease burden across a lifetime. Onset of hypertension during younger age is known to lead to higher cardiovascular morbidity, as well as overall cumulative mortality [26]. The new millennium has brought us significant improvements in healthcare access and resources, and therefore it is disappointing, even devastating, that some of the global leaders in healthcare, such as the United States, have reported a significant increase of 34.2% of deaths directly attributed to hypertension between 2009 and 2019 [27].

Inequity regarding epidemiological distribution between genders seems to be obvious. According to standardized blood pressure cut-off values used to diagnose hypertension, the overall burden of hypertension prevalence affects males in higher proportion. Nevertheless, it should not be neglected or forgotten that females are at higher risk of developing cardiovascular complications at lower blood pressure thresholds, and that the impact of hypertension is not uniform between the sexes [28].

Gender-related differences in hypertension range from simple to complex, reflecting a sophisticated molecular mechanism. Blood pressure control mechanisms are dependable on various gender specific differences, such as the activity of sympathetic nervous system, angiotensin-converting enzyme, endothelia-1, and sex hormones [29]. Furthermore, starting from adolescence, men have higher blood pressure values compared to women, and these differences persist averagely throughout the sixth decade of life [30].

Sex hormones, especially estrogen and testosterone, seem to be a key link and the chain of blood pressure controls the differences between the genders. Aside from maintaining blood pressure control and modifying functions of the renal, central, and vascular system, they affect numerous other pathways linked to the control of blood pressure [30,31]. Estrogen is an antihypertensive sex hormone; it can limit blood pressure rising, and it has cardioprotective properties, while androgens appear to have some pro-hypertensive as well as some anti-hypertensive features. Surgical removal of androgen sources leads to blood pressure decrease, as is confirmed in animal models [32]. On the other hand, paradoxically, deficiency of androgens might be associated with increased risk of hypertension development [31].

During their lifetime, women are facing the risk of developing two distinct, gender-specific types of hypertension: postmenopausal hypertension and hypertensive disorder of pregnancy (HDP). Postmenopausal hypertension is accompanied by estrogen depletion, and aside from changes in molecular pathways leading to vasoconstriction, interference of other factors associated with menopause, such as depression and anxiety, may have an additional impact on postmenopausal hypertension development [33]. HDP is the leading cause of maternal and fetal morbidly and mortality [34], and will be further elaborated in more detail in this review. Another entity of hypertension that displays a difference in manifestation frequency between the genders is the white coat hypertension (WCH) phenomenon. The WCH phenomenon is more commonly attributed to females, and therefore women require more thorough inquiry as well as proper monitoring and surveillance in order to establish adequate diagnosis and to pursue the most appropriate mode of treatment [35].

Hypertension is in most cases a controllable condition; broad therapeutic approaches are applicable as treatment options, but disappointingly low awareness of the magnitude of the consequences accompanied by this serious condition seems to be the key obstacle in reducing the rate of hypertension and the consequences associated with it [25].

## 4. Stroke and Pregnancy

Pregnancy and puerperium are known to be the state of disturbed hematological mechanisms resulting in pro-thrombotic inclination [36]. Ischemic stroke, ischemic heart disease and venous thromboembolism underline thrombosis as their predecessor [37]. Ischemic stroke and ischemic heart disease are accountable for one in four deaths globally [38] and thromboembolic disease is the leading cause of maternal mortality [36]. The peripartum and postpartum period carry the highest stoke risk, and it is estimated that stroke affects approximately 30 in 100,000 pregnancies [39]. High-risk pregnancies, particularly pregnancies carrying the burden of HDP, have up to a six-fold higher chance of suffering from this serious and potentially even fatal condition [39]. Disturbingly, up to 1 in 500 women facing preeclampsia are at risk of acute cerebrovascular disorder [40]. Stroke in pregnancy is a rising concern worldwide [41]. A study conducted in Canada revealed a concerning, even dramatic, 60% increase in stroke incidence during pregnancy in a 13-year period, between 2003 and 2016 [41]. Even though pregnancy and peripartum strokes are in general rare, they require special attention, due to the specificity of clinical presentation and significant maternal mortality [2].

Ischemic stroke manifestations in pregnancy do not differ significantly from those of non-pregnant women and men. They are usually presented as focal neurological abnormalities and are dependable on the localization of vascular lesion. Most common presentations include weakness, cranial nerve abnormalities, and sensory changes [42]. In a case of hemorrhagic stroke, such as subarachnoid hemorrhage, patients main complain of an intense headache, usually described as the worst headache of their life. Other commonly observed symptoms may involve decreased consciousness, nausea, vomiting, neck stiffness, seizures, and focal neurological abnormalities [42]. Cerebral venous thrombosis is another significant cause of stroke, primarily associated with venous stasis and hypercoagulopathy, as the pregnancy itself is. Usual clinical manifestation is a severe headache accompanied with nausea and vomiting, papilledema, and other signs of increased intracranial pressure [42].

Diagnosis of stroke should be established as soon as possible after the onset of symptoms, and it requires further diagnostics, including cardiac investigation, extracranial vessels examination, and thrombophilia screening [2]. Clinical presentation and physical examination are usually insufficient to provide diagnosis, and especially to distinguish whether it is a case of ischemic or hemorrhagic stroke. The preferred imaging tool is magnetic resonance, but in the setting of limited resources, computer tomography should not be avoided, since the benefit of accurate diagnosis overweighs the risk of potential teratogenic effect [43]. Cerebral venous thrombosis, ischemic, and hemorrhagic stroke have an approximately equal etiology percentage share in stroke incidence during pregnancy [2].

A great variety of conditions, primarily cardiovascular, is associated with increased risk for ischemic stroke. Paradoxical embolism, cardiomyopathy, arterial dissections, posterior reversible encephalopathy, and reversible cerebral vasoconstriction are the most common ones [43]. Aside from cardiovascular disease, diabetes, sickle cell disease, thrombophilia, and substance abuse are all very well-recognized risk factors [43].

The risk of pregnancy-related stroke is increased among women aged 35 or more [43]. Advanced maternal age and a raising trend of delayed motherhood are significant contributors of overall pregnancy-related morbidity [44,45,46]; however, it remains difficult to explain such a dramatic increase in pregnancy-related stroke solely based on these criteria.

Lanska et Kryscio analyzed stroke and intracranial venous thrombosis risk factors among nearly 1.5 million deliveries and they identified HDP and caesarean delivery as the most prominent risk factors [47]. Furthermore, the same research duo found significant association between stroke and electrolyte and acid-base disorders, as well as between infection, excluding influenza and pneumonia, with intracranial venous thrombosis [47]. Among patients of younger age, infections are recognized as a possible trigger for stroke. Sebastian et al. studied association of infections and stroke in the general population and found that all types of infections are associated with increased stroke incidence, and emphasized the impact of urinary tract infections [48]. Attention to infection being a possible trigger for stroke was driven by Miller et al. as well, who anticipated the association of stroke during delivery and the presence of infection at hospital admission [49]. In the study analyzing the association between infection during delivery and readmission risk within 30 days of delivery due to postpartum stroke, Miller et al. found positive association for ischemic, but not for hemorrhagic stroke [50].

Hypertension is undoubtedly a major risk factor associated with stroke [51]. Preeclampsia is part of the HDP spectrum, and it displays similar pathophysiological patterns and overlaps with stroke. Hypertension, endothelial dysfunction, dyslipidemia, hypercoagulability, and cerebral vasomotor reactivity abnormalities are common denominators for both of these conditions [52]. Aside from well-known pregnancy- and puerperium-related risk, preeclampsia displays a lifelong risk for several other conditions that may eventually lead to various health disorders and even premature death. Furthermore, preeclampsia is undoubtedly recognized as a lifelong risk factor for fatal and non-fatal stroke [52].

Due to its complexity, preeclampsia requires multidisciplinary team efforts, gathered around obstetricians and cardiologists, in order to decrease its negative long-term impact on cardiovascular function and women’s health [53]. Ultimately, the complexity of preeclampsia poses another significant question, the health and the wellbeing of the offspring. It is the matter of raising the concern of how and in what way does it impact the offspring [54] and should some specific measures of surveillance be provided for these children during their growth and later during adulthood.

Cardiovascular disease, hypertension, diabetes, dyslipidemia, and hypercoagulability [43,44] are among the leaders of pregnancy-related stroke. Most of these risk factors are modifiable, and most importantly preventable. Monitoring and control of these risk factors should provide us a potent platform designated to decrease stroke incidence.

## 5. Hypertensive Disorder of Pregnancy (HDP)

There are four distinct entities of hypertensive disorder of pregnancy (HDP) which are gathered under a common denominator, blood control failure (Figure 1). They include gestational hypertension, chronic hypertension, preeclampsia/eclampsia, and preeclampsia superimposed on chronic hypertension [55,56]. Gestational hypertension is diagnosed after 20 weeks of pregnancy as systolic and diastolic blood pressure values higher than 140 mmHg and 90 mmHg, respectively, in two separate measurements, taken four or more hours apart [56,57]. Chronic hypertension is diagnosed before the 20th week of gestation, and when complicated with preeclampsia, it leads to another, separate HDP entity—preeclampsia superimposed on chronic hypertension [55]. Diagnosis of preeclampsia is most commonly made in the presence of hypertension and proteinuria (at least 300 mg of proteins in 24 h urine collection). Additionally, in the absence of proteinuria, diagnosis can be established if a patient presents with new-onset thrombocytopenia, impaired liver function and/or epigastric pain or right upper quadrant pain, renal insufficiency, pulmonary edema, headache, or vision problems [56].

Preeclampsia complicates up to 8% of all pregnancies globally and is accountable annually for nearly 50,000 maternal deaths and approximately 500,000 newborn and fetal deaths worldwide [56,57]. Preeclampsia is a progressive, multi-system disorder caused by placental malperfusion and it is associated with the release of multiple soluble factors into maternal circulation, resulting in generalized vasoconstriction, endothelial dysfunction, and systemic inflammation [58]. These soluble factors were proposed as the most promising biomarkers for the early discovery and diagnosis of preeclampsia, and they include placental growth actor (PlGF), soluble fms-like tyrosine kinase-1 (sFLT1), vascular endothelial growth factor (VEGF), and soluble endoglin (sENG) [59]. However, meta-analysis has revealed that the accuracy and reliability of these four biomarkers in clinical practice are poor for precise preeclampsia prediction [57].

Patients suffering from preexisting hypertension, insulin-dependent diabetes mellitus, chronic kidney disease, as well as patients with a positive history of early-onset preeclampsia, are considered high-risk patients [60]. Other common risk factors include age older than 40, obesity, pregnancy resulting from donor egg cell, donor insemination, in vitro fertilization, multiple pregnancies, positive family history of preeclampsia, and antiphospholipid syndrome [61]. Research suggests that ethnical and racial background may also have a role in preeclampsia risk stratification, but these differences are complex and can be influenced by a variety of factors. Non-Hispanic Black women, American Indian, and Alaskan Native women are found to have a higher incidence of preeclampsia compared to women of European descent. In addition, Hispanic and Asian population were found to be a group of increased risk compared to European population [62].

Understanding the pathophysiology of preeclampsia was a long and complex journey. First, it was considered to be a central nervous system disease, manifested with the presence of seizures, but is considered today to be primarily a vascular disorder [61]. Progression of preeclampsia may lead to eclampsia, manifested with generalized tonic-clonic seizures [63]. Lisonkova et al. analyzed the relationship between preeclampsia and eclampsia among more than a million singleton pregnancies during a four-year timeframe. According to their study results, eclampsia risk was higher at term, and the risk for severe preeclampsia declined near term [64].

Preeclampsia unfortunately remains underestimated as a risk factor for future perimenopausal and postmenopausal disease, including cerebrovascular, cardiovascular, kidney, and liver diseases. Obesity, diabetes, hypercoagulable state, dyslipidemia, and endothelial dysfunction are known to be factors of increased risk for both of these conditions, and therefore it is debatable whether preeclampsia is a trigger for these diseases by itself, or whether it is simply an early marker for women at increased risk [65].

HDP is one of the major contributors of iatrogenic prematurity, and according to projections coming from Kuan Wang et al., when accompanied with preterm delivery, history of HDP increases the risk for subsequent stroke in women by 3.22-fold [66].

## 6. Placental and Cerebral Malperfusion: Clinical Manifestations and Differential Diagnosis

Pregnancy imposes significant vascular changes and adaptations, closely related to the development of the placenta, an organ that is similar to preeclampsia and exclusively associated with the pregnancy. Placenta has the central role in the promotion of fetal and maternal health, and along with fetal growth restriction, stillbirth, and recurrent pregnancy losses, preeclampsia is one of the main complications resulting from the impaired development of the placenta [67]. Abnormal trophoblast invasion of the spiral arteries leads to inadequate uteroplacental blood perfusion, which subsequently promotes oxidative stress reactions in the placenta, as well as a further interception with normal trophoblast invasion and placental angiogenesis [68]. Immunological patterns explaining preeclampsia progression rely on disturbances related to maternal immunological adaptation to the trophoblast, which consequently leads to inadequate perfusion and hypoxia of the trophoblast. Hypoxia damages the trophoblast tissue, and the reperfusion amplifies the process. Thereby, the maternal inflammatory system is repeatedly activated, and cellular apoptosis is accelerated, leading to placentation disorders, imbalance between pro- and anti-angiogenic factors, and endothelial dysfunction [69]. As we have argued, among biomolecules involved in early placental vasculogenesis, VEGF, PlGF, sFlt-1, and s-Eng have been identified as triggering factors for abnormal placentation and the early biomarkers of the disease [70], but novel research has shifted its focus of interest more dominantly to genetic markers of preeclampsia. Wang et al. have recently identified three genes associated with placentation as potential genetic biomarkers that might revolutionize prediction, and more importantly treatment of preeclampsia [71].

Cerebral circulation has several peculiarities, including integrity of the blood-brain barrier and cerebral autoregulation. In normal, normotensive pregnancies, cerebral arteries undergo a remodeling process resulting in increased capillary density and improved autoregulation [72]. In the setting of preeclampsia, loss of blood-brain-barrier integrity and cerebral autoregulation leads to neurovascular disfunction, lower cerebrovascular resistance, higher baseline cerebral perfusion pressure, and decreased vasodilatation [39]. Neurovascular disfunction ultimately leads to the breakdown of the blood-brain barrier and vasculopathy, findings that were confirmed on postmortem neuropathological enquires decades ago [73].

Resemblance shared between placental and cerebral vascular changes, adaptations, and sophisticated auto-regulatory mechanisms are not merely coincidental; they reflect distinctive and complex cardiovascular performances occurring in the maternal circulatory system. In the setting of preeclampsia, difficulties regarding the maintenance of these cardiovascular performances remain even more challenging. Furthermore, suboptimal performance of cardiovascular systems is nowadays proposed as a more likely cause of uteroplacental malperfusion, leading us to assume that placental dysfunction is a secondary condition initiated by the primary disturbances of cardiovascular system, rather than a primary immunological disorder associated with abnormal trophoblast invasion. Studies focusing on maternal echocardiography and early angiogenic biomarkers have revealed that changes in these parameters can be observed weeks, or even months, before preeclampsia develops [74]. Malperfusion seems to be the cornerstone related to both of these conditions; placental malperfusion leads to preeclampsia, and cerebral to stoke [38,74].

Complex pathophysiological mechanism leading to preeclampsia have resulted in the heterogeneity of clinical signs and symptoms of this disease. In some cases, it is exceptionally difficult to provide a proper, timely diagnose, and to distinguish stroke from eclamptic seizure. Preeclampsia is associated with multiple organ disfunction; its signs and symptoms are mainly associated with proteinuria and/or end-organ damage and are accompanied by abnormalities in laboratory findings. The most common symptoms include sudden swelling of the face, hands, and feet, epigastric or right upper quadrant pain, signs of impaired liver or kidney function, thrombocytopenia, and pulmonary edema [75,76]. Neurological complications associated with preeclampsia involve headache, vision problems (blaring or flashing), cerebral edema, and seizures, but acute cerebrovascular accidents may occur as well [77]. Headache, unilateral face and/or limb weakness, confusion, difficulty speaking or understanding, dizziness, sudden trouble walking, or balance loss are the usual signs and symptoms of stroke. Typically, they evolve rapidly and occur suddenly [78].

Headache is the most common symptom, and it is mutual for both of these conditions. Headache often accompanies puerperium, and it is one of the main complaints that peripartum women experience. Peripartum procedures, changes in hormonal status, and physiological changes are the most common causes of headache, but it must not be forgotten that it can also be associated with serious, life-threatening conditions, including preeclampsia or stroke. In order to differentiate conditions and causes associated with headache, clinical evaluation of headaches should be carefully conducted [79]. It should be stressed and never forgotten that primary headaches, including tension-type headaches and migraines, per se carry an increased risk for premature delivery and poor pregnancy outcome, regardless of the presence of preeclampsia [80]. Surprisingly, in eclampsia, headache may be absent in up to 50% of women before the onset of ecliptic seizures, and therefore it has unsatisfactory predictive performance for eclampsia [81]. Unlike migraines, where headache tends to be more dominant on one side of the head, in preeclampsia and eclampsia, headache is not particularly distinctive—it is usually holocephalic, located all over the head [82]. In stroke, headache is usually located in the affected area related to the stroke localization, it is usually acute in the onset, and it peaks rapidly. Back-of-head headache is most commonly triggered by a blockage of the blood vessels providing vascularization to the back of the brain, while forehead headaches are usually associated with blockades in carotid arteries. Headache caused by a stroke is usually not accompanied by any other sign or symptom [83]. Focal neurological symptoms are a more common stroke manifestation among patients suffering from hemorrhagic compared to ischemic stroke [84]. Pregnancy and the early postpartum period are delicate and usually the most vulnerable parts of a woman’s life. Among women faced with previous cardiovascular, hematological, or neurological conditions, it is the time of possible overlaps between new onset headaches and the increased risk of stroke, and therefore these patients require special attention [83]. Since preeclampsia is accompanied with specific, pregnancy-related features, obstetricians have the possibility to distinguish some pregnancy-related features of preeclampsia from stroke. Fetal ultrasonography is a valuable diagnostic tool; it provides prompt assessment of the fetal condition, and it reveals the fetal conditions commonly associated with preeclampsia, such as fetal growth restriction [85].

## 7. Therapy and Pregnancy

In most cases, with properly and timely applied measures of prevention, stroke is predictable, and preeclampsia is a controllable condition. Risk identification and stratification, timely intervention, and good patient compliance are therefore key links in the chain of preeclampsia and stroke prevention (Figure 2) [58,86,87].

Stroke therapy is complex, and details of this multidisciplinary treatment exceed the scope and the content of this review. Acute stroke reperfusion therapy is the treatment of choice in cases of ischemic strokes, but unfortunately a great majority of pregnant, as well as postpartum, women do not receive this type of treatment, even though pregnant and postpartum patients have similar favorable short-term outcomes as their non-pregnant peers [88]. Recombinant tissue plasminogen activator (rtPA) is given routinely to non-pregnant patients presenting with ischemic stroke within 3–4.5 h from the stroke onset. rtPA reduces morbidity and overall mortality risk within 90 days of stroke, but primarily due to pregnancy or recent surgery, pregnant and postpartum women are receiving this treatment modality less frequently [42,88].

The American College of Obstetrics and Gynecology (ACOG) recommends prophylactic use of low-dose aspirin for all women facing the risk of preeclampsia. Daily use of 81 mg of aspirin is a recommendable dose and it should be initiated between the 12th and 28th gestational week, preferable before the 16th gestational week, and it should not be ceased until the delivery [56]. Even though systemic reviews and meta-analysis have confirmed that low-dose aspirin (75–81 mg) is effective in reducing risk for term preeclampsia, higher doses of aspirin (150–162 mg) are superior in preventing term as well as preterm preeclampsia [89]. In cases when high doses of aspirin are indicated, the International Federation of Gynaecology and Obstetrics (FIGO) recommends that therapy should be ceased at 36th gestational weeks [90]. Furthermore, aspirin use has demonstrated its effectiveness in reducing long-term stroke risk among women with prior HDP [91]. Magnesium sulfate is recommended for high-risk patients, women facing eclampsia, as well as preeclampsia and gestational hypertension with severe features. Magnesium sulfate should be continued up until 24 h following the delivery, and it is intended to help in the prevention, as well as the management, of seizures during the stabilization time [92]. Tendon reflex, respiration, and urinary output ought to be mandatorily monitored [93]. Calcium supplementation of 1 g daily starting from the 12th gestational week is recommendable for patients at risk for preeclampsia and in cases of low-calcium diets [94].

Stroke, myocardial ischemia, congestive heart, and renal failure are potentially serious complications of severe hypertension lasting for more than 15 min [88]. Oral administration of nifedipine or intravenous administration of labetalol or hydralazine should be introduced as soon as possible to avoid this serious complication [88,95]. Even though preeclampsia and its systemic manifestations can be controllable, delivery of the placenta remains a definitive treatment for preeclampsia [57]. Dosage, effects, and comments for elaborated therapeutic agents are presented in Table 1.

## 8. Conclusions

In conclusion, understanding the differences between preeclampsia and stroke in pregnancy is vital for healthcare providers to enhance clinical decision-making, improve patient care, and promote positive pregnancy outcomes. The management approaches for preeclampsia and stroke require a multidisciplinary approach involving obstetricians, neurologists, and other healthcare professionals. Timely diagnosis, risk assessment, appropriate interventions, as well as special caution of pregnancy-related hypertension management are crucial in optimizing maternal and fetal outcomes. Although preeclampsia and stroke in pregnancy present unique challenges, early recognition, regular antenatal care, and close monitoring can contribute significantly to the prevention, early intervention, and management of these conditions.

Further research and collaboration between obstetricians and neurologists are warranted to advance our knowledge and develop targeted interventions for these complex conditions.

## Figures and Tables

**Figure 1 medicina-59-01707-f001:**
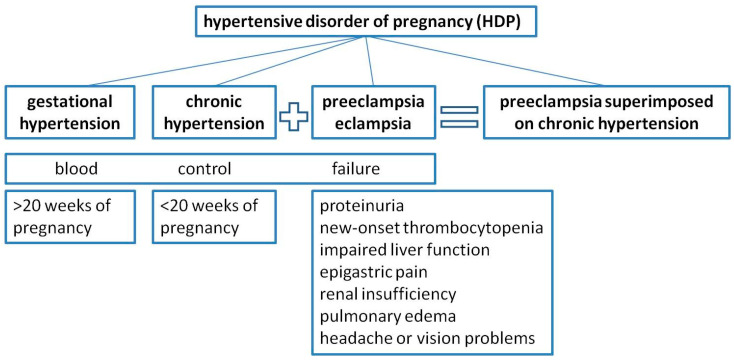
Classification and main characteristics of hypertensive disorder of pregnancy (HDP).

**Figure 2 medicina-59-01707-f002:**
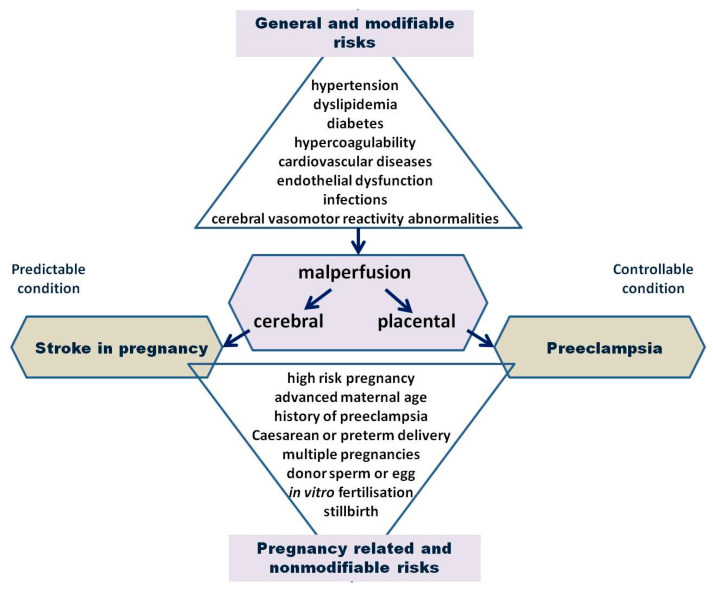
The pathophysiological patterns and major risk factors associated with stroke in pregnancy and preeclampsia.

**Table 1 medicina-59-01707-t001:** Overview of therapeutic agents used for preeclampsia prevention and treatment.

Therapeutic Agents	Dosage	Effects	Comment
Aspirin	75–81 mg [56] 150–162 mg [89]	Reduces the risk of term preeclampsia [89] Reduces the risk of preterm and term preeclampsia [89]	Continued until delivery [56] Continued until 36th gestational week [89]
Antihypertensive drugs: 1.Labetalol 2. Hydralazine 3. Nifedipine	10–20 mg intravenously, followed by 20–80 mg every 10–30 min or constant infusion 1–2 mg per minute intravenously. Maximum cumulative dosage is 300 mg [56] 5 mg intravenously or intramuscularly followed by 5–10 mg intravenously every 20–40 min or constant infusion of 0.5–10 mg per hour. Maximum cumulative dosage is 20 mg [56] 10–20 mg orally. When needed it can be repeated in 20 min. Daily dosage is 10–20 mg every 2–6 h to a maximum dose of 180 mg [56]	Onset of action is 1–2 min [56] Onset of action is 10–20 min [56] Onset of action is 5–10 min [56]	Possible side effects include headache, reflex tachycardia, fewer and abnormal fetal heart rate. Labetalol should be avoided in in women with asthma and preexisting cardiac diseases [56]
Magnesium sulfate	Intravenous loading dose of 4–6 g over 20–30 min followed by maintenance dose of 1–2 g hourly [56]	Intended to help in prevention, as well as the management, of seizures during stabilization time [92]	Infusion should continue for 24 h after delivery. Deep tendon reflexes are lost at magnesium serum levels of 9 mg/dl, respiratory distress occurs at 12 mg/dl and cardiac arrest at 30 mg/dl [93]
Calcium supplementation	1 g daily starting from the 12th gestational week [94]	Reduces preeclampsia risk for patients at high risk and in cases of low-calcium level diets [94]	/

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
