# Peer review of "Stroke vs. Preeclampsia: Dangerous Liaisons of Hypertension and Pregnancy"

_medicina, 2023, doi:10.3390/medicina59101707_

Round 1

Reviewer 1 Report

the Abstract is too long. is not clearlly specified the type of study, matherial and methods. - must be reorganized

The Review plan must be restructured 

it looks like a book chapter- more images (physiopathology - graphical representation!

the first 3 chapter are too  general 

some of the references are too old

 For the Stroke in pregnancy- tabel with all studies ( limited period of time) which describe these issues ( hemoragic/ ischaemic)

a graphycal scheme of preclampsia biomarkers/ cerebral flow

tabel with the treatment of hypertension in pregnancy

-

Author Response

Thank you for your valuable comments and suggestions.

1) the Abstract is too long. is not clearlly specified the type of study, matherial and methods. - must be reorganized

- We adjusted the length of abstract and provided clarification regarding the type of study, however since this is not a systematic review we could not implement your suggestion regarding the material and methods

2) The Review plan must be restructured 

- We feel confident that our study design follows the conceptualisation of a review paper and that it is adjusted for target readers for better clarification and understanding of the topic

3) it looks like a book chapter- more images (physiopathology - graphical representation!

- We inserted the addition graphic and table according to your suggestions

4) the first 3 chapter are too  general 

- To the best of our knowledge we have provided to our readers necessary facts  for introduction and better understanding of such a complex topic

5) some of the references are too old

- Majority of the references are within 10 years. However, some references are older, but needed to be mentioned due to the significance and importance of these discoveries

 6) For the Stroke in pregnancy- tabel with all studies ( limited period of time) which describe these issues ( hemoragic/ ischaemic)

- In the last 10 years only in PubMed data base, according to suggested search string for key word stroke (all fields) and pregnancy (all fields) we found 2396 articles, aside Cochran, Scopus and EMBASE. Therefore we suggest future research in form of systematic review and meta-analysis.   

7) a graphycal scheme of preclampsia biomarkers/ cerebral flow

-As we have discussed in our review novel research has shifted its focus of interest more dominantly to genetic markers of preeclampsia, and therefore we have not discussed biomarkers of preeclampsia in so many details.

8) tabel with the treatment of hypertension in pregnancy

-We have added a table with hypertension treatment as you have requested.

Reviewer 2 Report

This a well written comprehensive review succeeding to provide a synthesis of the existing literature on the clinical manifestation and differences between stroke in pregnancy and preeclampsia.
Authors should review the article for the appropriate use of the words Genders (Men, Women) or sexes (male, female) depending on the used reference.
Page 5: cardiomyopathy instead of cardiomyopathy
Page 7, Line 1-2: “Preeclampsia complicates up to 8% of all pregnancies and is accountable annually for nearly 50,000 maternal deaths and approximately 500,000 newborn and fetal deaths” You should report the population in which the reported numbers and percentages referred to (worldwide, U.S.A., Europe?)
Page 10, Line 14: “before the 16th gestational weak” correct it with “before the 16th gestational week”

Author Response

Dear Reviewer,

Thank you for your valuable comments and suggestions.

  • Authors should review the article for the appropriate use of the words Genders (Men, Women) or sexes (male, female) depending on the used reference.

- We have implemented changes according to your suggestions for genders and sex.

2) Page 5: cardiomyopathy instead of cardiomyopathy

- Thank you four your comment, we have corrected this mistake.

3) Page 7, Line 1-2: “Preeclampsia complicates up to 8% of all pregnancies and is accountable annually for nearly 50,000 maternal deaths and approximately 500,000 newborn and fetal deaths” You should report the population in which the reported numbers and percentages referred to (worldwide, U.S.A., Europe?)

- Thank you for your comment, we have specified the population in question.

4) Page 10, Line 14: “before the 16th gestational weak” correct it with “before the 16th gestational week”

- Thank you for your comment, we have corrected this mistake.

Kind regards,

Authors

Reviewer 3 Report

This is an important and excellent review. Are there racial differences in the risks and incidence of pre-eclampsia? What is the argument for 160mg dose of low dose aspirin? When does aspirin is stopped - at delivery or 36 weeks.

Author Response

Dear Reviewer,

Thank you for your valuable comments and suggestions.

1) Are there racial differences in the risks and incidence of pre-eclampsia?

- According to your suggestions, we have added this important risk factor for preeclampsia development. Thank you.

2) What is the argument for 160mg dose of low dose aspirin? When does aspirin is stopped - at delivery or 36 weeks.

- We have provided additional argumentations for low and high dose aspirin intake as you have suggested. Thank you.

Kind regards,

Authors